# The Influence of School Backpack Load on Dynamic Gait Parameters in 7-Year-Old Boys and Girls

**DOI:** 10.3390/s25134219

**Published:** 2025-07-06

**Authors:** Paulina Tomal, Anna Fryzowicz, Jarosław Kabaciński, Dominika Witt, Przemysław Lisiński, Lechosław B. Dworak

**Affiliations:** 1Department of Physiotherapy, Karol Marcinkowski Poznan University of Medical Sciences, 61-545 Poznan, Poland; 2Department of Biomechanics, Poznan University of Physical Education, 61-871 Poznan, Poland; fryzowicz@awf.poznan.pl (A.F.); kabacinski@awf.poznan.pl (J.K.); 3Department of Rehabilitation and Physiotherapy, Poznan University of Medical Sciences, 60-545 Poznan, Poland; dwitt@ump.edu.pl (D.W.); plisinski@ump.edu.pl (P.L.); 4Faculty of Health Sciences, Calisia University, 62-800 Kalisz, Poland; biomechanika-dworak@wp.pl

**Keywords:** school backpack, gait analysis, plantar pressure, symmetry, biomechanics, children

## Abstract

School-aged children are routinely exposed to additional physical stress due to carrying school backpacks. These backpacks often exceed recommended limits and can contain not only books and notebooks but also laptops, water bottles, and other personal items. The present study aimed to evaluate the impact of different backpack loads (10%, 15%, and 20% of body weight) on dynamic gait parameters in 7-year-old girls and boys. Twenty-six children (13 girls, 13 boys) participated in the study. Gait analysis was performed using the Footscan® system (RSscan International, Olen, Belgium; 2 m × 0.4 m × 0.02 m, 16,384 sensors) equipped with Footscan software version 7 (Gait 2nd generation), examining peak force (FMAX), peak pressure (PMAX), contact area (CA), and time to peak force (Time to FMAX) across five anatomical foot zones. The study revealed significant changes in all parameters, particularly at loads of 15% and 20% of body weight. Increases in plantar pressure, contact area, and asymmetry were observed, along with delays in time to peak force. These findings support the recommendation that children’s backpack loads should not exceed 10% of their body weight to prevent potential adverse effects on postural and musculoskeletal development.

## 1. Introduction

The early school years mark a critical period of development during which children begin to regularly carry school backpacks. These loads often include not only textbooks and notebooks but also items for extracurricular activities, such as sports uniforms and electronic devices. As educational demands increase, the size and weight of school supplies also grow, contributing to heavier backpacks.

The use of backpacks remains the most common method for children to transport school materials [1]. During the school day, children often walk with backpacks from one classroom to another. In relation to their body weight, young schoolchildren are particularly vulnerable to overload. Despite numerous ergonomic recommendations, excessive backpack loads continue to be a concern.

Although multiple studies have explored the impact of backpack carriage on posture, gait, and muscle activity, most of this research focuses on adults. There is a lack of consensus on safe weight limits for schoolchildren. Government and professional organizations have suggested various guidelines. International bodies like the AOTA, NSC, and CPTA suggest limits ranging from 10% to 15% of body weight [2].

Nonetheless, research shows that a significant number of children exceed these recommendations [3]. According to the Central Institute for Labour Protection, nearly 44.2% of urban children and 37.7% of rural children carry overweight backpacks [4].

Children exhibited a slower walking speed under additional load. A reduction in step length and single support time was observed, while their base of support width, step time, and double support time increased. No significant changes in stride length or gait cycle duration were found. The most pronounced kinematic alterations were noted at loading levels of 10% body weight (BW) and above [5].

The first changes in plantar pressure distribution are observed within a few weeks of children beginning to walk independently. Notable increases in both the contact area of the foot and the maximum force have been reported. These changes are believed to result primarily from the morphological development of the child’s foot [6]. Pau et al. did not find any statistically significant differences between limbs and therefore treated the right and left limbs together. However, Mickle et al. reported different values between the limbs and chose to analyze only one (either Right or Left) [7]. Pau et al. studied 218 children aged 6–13 years who carried school backpacks weighing an average of 5.2 kg—more than 15% BW in over half of the participants. During the test, children walked a distance of 8 m at a self-selected speed. The authors divided the foot into three zones: forefoot, midfoot, and rearfoot. The added school backpack load led to increased pressure in the forefoot by 25%, in the midfoot by 19.2%, and in the rearfoot by 8.1% [8]. Similar trends were observed in the present study, with pressure changes also noted in these three anatomical regions.

Plantar pressure refers to the distribution of pressure exerted between the plantar surface of the foot and the supporting substrate during functional locomotor tasks. The data obtained from plantar pressure analysis are of significant relevance in gait and postural studies, serving as a critical tool in the assessment of lower limb pathologies, the design and evaluation of footwear, sports biomechanics, injury prevention strategies, and various clinical and ergonomic applications [9]. Recent epidemiological studies indicate a rising prevalence of musculoskeletal disorders among school-aged children. Postural abnormalities, back pain, and early signs of biomechanical overload have been reported in up to 30–50% of children, with an increasing incidence correlated with age and load exposure. Several studies have linked an excessive school backpack weight to spinal deviations, altered gait mechanics, and fatigue-related complaints, underscoring the importance of preventive measures in this population [10,11]. 

There is a clear need to evaluate the biomechanical effects of backpack loads, especially in children. Understanding how different load levels affect gait and foot mechanics can inform preventive strategies and ergonomic guidelines. This study aims to analyze dynamic gait parameters under varying backpack load conditions in children aged 7 years, assessing the extent of the asymmetry, pressure distribution, and mechanical adaptations that may influence long-term health outcomes.

## 2. Materials and Methods

### 2.1. Participants

The study included 26 children aged 7 years, comprising 13 girls and 13 boys, all attending the first grade of a randomly selected public primary school in Poznań, Poland. The sample size was calculated using G*Power software (version 3.1.9.7), based on an expected effect size of 0.4, a power of 0.80, and an alpha level of 0.05. This resulted in a minimum required sample of 24 participants. To strengthen the group balance and statistical reliability, the final sample included 26 children (13 males and 13 females). The inclusion criteria required the absence of postural abnormalities, musculoskeletal deformities, pain during locomotion, lower limb injuries, and neurological disorders. The participants were not professionally engaged in competitive sports, including football. Participation was contingent on informed consent being provided by a parent or legal guardian. The study protocol was approved by the Bioethics Committee of the Poznan University of Medical Sciences and was conducted with the consent of school administrators and teaching staff.

### 2.2. Anthropometric Data

The basic anthropometric characteristics of the study group are presented in Table 1. The mean body mass was 27.84 ± 7.35 kg, the mean height was 126.5 ± 7.12 cm, and the mean BMI was 17.15 ± 2.8 kg/m^2^. 

### 2.3. Backpack Load Conditions

Three load conditions were tested, corresponding to 10%, 15%, and 20% of each child’s body weight (BW). The weight was adjusted using textbooks and notebooks packed into a standardized school backpack model (Herlitz Smart), which alone weighed 290 g. The final backpack weights, including the content, were 1.884 ± 0.73 kg (10% BW), 3.276 ± 1.1 kg (15% BW), and 4.668 ± 1.47 kg (20% BW).

### 2.4. Gait Measurement Protocol

Gait parameters were assessed using the Footscan® pressure platform (RSscan International; 2 m × 0.4 m × 0.02 m, 16,384 sensors), in conjunction with Footscan 7 gait second-generation software. Children walked barefoot at a self-selected speed (Figure 1).Each test condition included five valid mid-gait trials, during which the child wore the backpack on both shoulders. The four analyzed conditions were:(1)barefoot walking without a load,(2)walking with 10% BW load,(3)walking with 15% BW load,(4)walking with 20% BW load.

**Figure 1 sensors-25-04219-f001:**
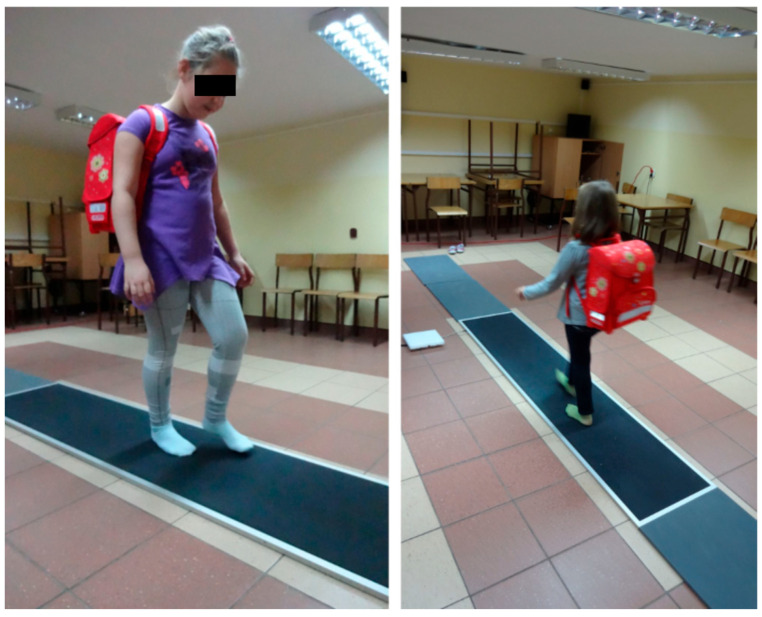
Measurement walkway, including level adjustments for surface alignment. Photograph by the author.

### 2.5. Parameters and Foot Zones

The following dynamic parameters were recorded:Peak vertical ground reaction force (FMAX) [N]Maximum plantar pressure (PMAX) [N/cm^2^]Foot-ground contact area (CA) [cm^2^]Time to peak force (Time to FMAX) [ms]

The plantar surface of the foot was divided into five anatomical zones for analysis: Hallux (big toe), Toes 2–5, Forefoot, Midfoot, and Heel. Data were analyzed separately for the left and right feet and stratified by sex.

### 2.6. Statistical Analysis

Descriptive statistics (mean ± standard deviation) were computed. The Shapiro–Wilk test (*p* < 0.05) assessed normality. Levene’s test verified variance homogeneity. Paired-sample *t*-tests or Wilcoxon tests were used for dependent variables; independent sample *t*-tests or Mann–Whitney U tests were used for between-group comparisons. Differences among the four loading conditions were examined using one-way ANOVA followed by Duncan’s post hoc tests (*p* < 0.05).

The Symmetry Index (SI) was calculated for each foot using the following formula:
SI = |(XR − XL)/XL| × 100%

where XR and XL represent the values from the right and left limbs, respectively. An SI ≥ 100% was considered indicative of pronounced asymmetry.

## 3. Results

### 3.1. Peak Vertical Ground Reaction Force (FMAX/PVGRF)

Significant differences were observed between left and right feet, particularly in the Hallux and Heel regions. For girls, FMAX increased significantly in the Hallux at 15% BW (*p* = 0.000068) and in the Heel at 20% BW (*p* = 0.012576). Among boys, the highest asymmetry occurred in the Toes 2–5 and Midfoot zones, with a Symmetry Index (SI) of 90.3% and >100%, respectively. This suggests a considerable imbalance in force distribution between limbs (Figure 2). 

### 3.2. Maximum Plantar Pressure (PMAX/MPP)

Increased plantar pressures were observed in the Forefoot and Midfoot zones. In girls, PMAX significantly increased in the Forefoot at 15% BW (*p* = 0.000290) and in the Midfoot at 20% BW (*p* = 0.000714). Boys showed elevated PMAX across the same zones, including the Heel, under heavier loads. Asymmetry was most pronounced in the Midfoot region with SI >100% (Figure 3).

### 3.3. Contact Area (CA/FGCA)

The contact area significantly increased with load, especially in the Midfoot and Heel. Girls showed a 25.3% increase in CA at the left Midfoot under 20% BW. Boys exhibited increases of 36.8% (left) and 21.6% (right) under the same condition. These changes reflect morphological adaptations that redistribute load and maintain balance (Figure 4).

### 3.4. Time to Peak Force (Time to FMAX/T)

Time to FMAX was delayed across most anatomical zones as the backpack load increased. Among girls, significant delays were recorded in the Hallux and Toes 2–5 zones, especially at 15% and 20% BW. For boys, delays were prominent in the Forefoot and Midfoot. These adaptations suggest modified loading patterns and gait rhythms due to weight-induced compensations (Figure 5).

## 4. Discussion

The most important findings from this study are as follows: FMAX and PMAX values increased progressively with heavier backpack loads.CA expanded most in the Midfoot and Heel zones, especially under 20% BW.Time to FMAX delays were greatest under 20% BW, indicating altered gait timing.

These findings underline the sensitivity of pediatric gait patterns to increased load and support, limiting the backpack weight to no more than 10% of body weight in young school children.

In the present study, a cautious approach was adopted by dividing the foot into five zones: the hallux (1st toe), toes 2–5, forefoot, midfoot, and heel. Although the pedobarographic platform Footscan used in the study allows for parameter analysis across ten distinct zones, the reduced size of children’s feet compared to adults could result in unreliable data when utilizing a more detailed zonal division. Bosch et al. opted for a simplified five-zone model [7,12].

The current study performed biomechanical analyses in 7-year-old children. To date, there is a paucity of research evaluating the impact of additional external load on gait in children of this age group.

Compared to adults, children exhibit lower peak plantar pressure values across all anatomical regions of the foot.

In the present study, the results were analyzed by sex. At the age of seven, children already show morphological and somatic differences. In the examined cohort, girls had a higher average body mass and height than boys. The mean differences were 2.7 kg and 1.92 cm, respectively.

When analyzing the differences between girls and boys in the present study, statistically significant differences were found in FMAX values for the hallux of the right limb under a 15% BW load and the left heel under a 20% BW load. Boys exhibited higher values than girls, by 13.6% and 19.3%, respectively.

For the Max P parameter, statistically significant differences were observed for the right heel and left toes 2–5 under a 15% BW load, as well as for the left midfoot under a 10% BW load. Boys again showed higher values—by 17.6%, 32%, and 27.6%, respectively.

In the group of boys, greater contact area values under the toes 2–5 zone were also recorded, both in the unloaded condition and under a 15% BW load.

The literature highlights sex-related differences in dynamic gait parameters. Unger et al. examined the plantar pressures in 42 children (20 boys and 22 girls) during their first year of bipedal locomotion. The initial measurement was taken at 16.1 months of age, with follow-up assessments every three months over a one-year period. They observed differences in dynamic load parameters and structural foot morphology. Boys exhibited wider midfeet and lower medial longitudinal arches. Greater dynamic loads, particularly in the forefoot and calcaneus regions, were observed in girls [13].

In this study, a “mid-gait” measurement protocol was applied. Based on pilot testing and the mature gait patterns observed in 7-year-old children, this method was deemed appropriate.

In the present study, an external load was introduced in the form of a school backpack weighing 10%, 15%, and 20% of body weight (BW). A single backpack model was used for consistency. The study was conducted under dynamic conditions, with children walking across a 2-m-long platform. Genitrini et al. found that Double Pack (DP) and T-pack (TP) designs demonstrate postural advantages. Compared to conventional backpack designs, DP provides superior outcomes in terms of balance and muscle activation; however, it is associated with certain limitations, including a restricted visual field, increased thermal sensation, and impaired ventilation [14].

In this study, the right (R) and left (L) lower limbs were analyzed separately. The objective was to verify the potential gait asymmetries observed under both unloaded and loaded conditions. Differences between the L and R limbs were identified in the analyzed kinematic and dynamic gait parameters.

The current study also reports the value of the gait symmetry index (WA), which is one of the most commonly used and referenced metrics for assessing asymmetries in this form of locomotion. A value of 0 indicates perfect symmetry, while a value of 100 or more suggests pronounced asymmetry [15].


**Maximum Vertical Ground Reaction Force (FMAX) [N]**


Data regarding dynamic (biomechanical) parameter changes in children under the influence of additional load from school backpacks remain limited. 

In the present study, the analysis of initial FMAX values—recorded under unloaded conditions—revealed statistically significant differences in the hallux zone between the left and right limbs in girls. The left side showed 29% higher force values (symmetry index: 34.5%). Among boys, the force was 33% higher on the left side (symmetry index: 39.6%).

Rodrigues et al., in a static study of 30 children aged 10 years, also reported approximately 6% higher force values on the left side by considering all toe zones combined [16].

Such asymmetry in loading may be partially attributed to limb dominance. It is generally accepted that the dominant limb initiates movement, while the non-dominant limb plays a greater supportive role [17]. The higher force values on the left side may indicate that this limb was acting as the primary support. However, the study did not include data on which limb initiated gait.

In the present study, when comparing changes in maximum force under applied loads, a decrease in FMAX was observed in the midfoot zone for both girls and boys under 15% and 20% BW loading on the left limb. Among girls, the force decreased by 55% and 53.8%, respectively. Among boys, the reduction was 47.8% and 54%.

These results are consistent with those reported by Rodrigues et al., who also found a decrease in FMAX in the right midfoot zone. In their study, a 4% decrease was already evident at a load of 5% BW, which was not tested in the current study. Rodrigues et al. also noted an increase in FMAX in the toe zone at 15% BW load. These findings partially align with the present results, which showed increased force in the toes 2–5 zone at both 15% and 20% BW. However, for the hallux zone at 20% BW in boys, a decrease in FMAX was observed.

This study also showed an increase in maximum force in the forefoot zone for both sexes and limbs under 15% and 20% BW loads. Rodrigues et al. reported a similar increase in FMAX for the right side and a decrease for the left; however, those values were not statistically significant [16].

The results demonstrated that the active force, braking force, impact peak, loading rate, active peak, maximum braking, hip flexion, and hip range of motion were all significantly elevated under load carriage conditions compared to unloaded walking. Conversely, the time to peak was reduced [18].


**Maximum Pressure (PMAX) [N/cm^2^]**


In the group of girls, the maximum pressure in the forefoot zone increased by 31.2% for the left limb and 25.4% for the right limb under a 15% BW load. In the left midfoot zone, pressure increased by 43.2% under a 20% BW load. For the heel zone, increases of 33.6% (right) and 27.3% (left) were recorded. Comparable values were obtained in the group of boys.

Drerup et al. examined the effects of external load on gait in 19 adults aged 34 years. Their findings indicated that the greatest pressure changes occurred in the midfoot and forefoot zones. The largest alterations were observed at a load of 20 kg, equivalent to approximately 26% BW of the participants [19].

Increased forefoot pressure may raise concerns, as this area is structurally composed of small bones with a limited capacity to dissipate the forces occurring during dynamic weight shifts.


**Contact Area (CA) [cm^2^]**


In girls, changes in the contact area were observed in the left midfoot zone, where CA increased by 25.3% under a 20% BW load. Among boys, changes were also noted in both the left and right midfoot zones, with increases of 36.8% and 21.6%, respectively, under the same load.

The results obtained in the present study are consistent with those reported by Pau et al., who found the most linear relationship between load and CA in the midfoot zone, and to a lesser extent in the forefoot. In their study, the observed changes reached up to 10%, but only for the highest tested load (>7 kg), and all measurements were conducted under static conditions [20].

In another study, Pau et al. examined the effects of school backpack use (average load of 5.2% BW) on CA during walking. The largest changes were observed in the midfoot (4.3%), followed by the forefoot (2.8%) and rearfoot (1%). They concluded that the impact of wearing a backpack on CA is more pronounced under static conditions than during walking [8].

Mickle et al. examined the feet of obese children and found that an increased body mass was associated with an increase in CA compared to children with a normal body weight. One of the main indicators of this change was a lowered medial longitudinal arch, which is likely due to sustained high loading over time. This scenario is typical in pediatric obesity [21].

Other researchers who investigated the effects of increased loading from body weight itself reached similar conclusions. In static conditions, obese children exhibited higher force values, greater CA and higher pressure values compared to children with a normal body mass. During walking, obese children also demonstrated an elevated plantar pressure, except in the toe zone. Despite the larger surface area for force distribution, plantar pressure in the midfoot and metatarsal head zones was markedly higher in obese children [22].


**Time to Reach Maximum Force (Time to FMAX) [ms]**


When analyzing the differences between girls and boys, statistically significant values were observed for the hallux zone on the right limb in unloaded conditions, with girls showing 4.8% longer times. The time to FMAX was prolonged in all foot zones except the right forefoot, in both sexes.

This parameter appears highly relevant, as it reflects changes in foot mechanics during gait in response to additional loading. Unfortunately, there is currently a lack of reference data for this measure, indicating the need for further research.

Comparing children with normal weight to obese children, time to FMAX was significantly longer under the toes 2–5 (left), hallux, toes 2–5 (right), the 4th and 5th metatarsal heads, and the midfoot of both limbs [23]. These findings suggest that the loading patterns in obese children mirror those seen during temporary loading from a school backpack.

Kinoshita also reported a prolonged time to FMAX, suggesting that it may result from an excessively long step (despite its overall shortening), where the heel contacts the ground too far behind the body’s center of mass in the sagittal plane. He concluded that such adaptations to additional load during gait serve to reduce the forces acting on the body and optimize movement [24].

Gait asymmetry was addressed by separately analyzing right and left limbs and calculating the symmetry index (WA). Asymmetry in the feet may be associated with scoliosis [15]. Habitual backpack use and an increasing backpack mass may contribute to the asymmetrical distribution of forces and plantar pressures. Early identification and intervention could help ensure proper pelvic and spinal alignment, particularly in scoliosis prevention.

An excessive schoolbag weight has been associated with increased physical stress and fatigue in students. The potential impact of habitual physical activity on gait kinematics and kinetics should also be considered when interpreting the results. In the present study, none of the participants were involved in systematic sports training or competitive athletic programs. Therefore, it is unlikely that extracurricular physical activity acted as a confounding factor influencing the observed effects of backpack load. Despite existing guidelines regarding backpack load, a substantial proportion of students in both urban and rural educational settings continue to carry schoolbags exceeding 10% of their body weight. These findings underscore the need to revise current standards and implement stricter regulations to ensure that schoolbag loads do not surpass the 10% body weight threshold in order to mitigate potential health risks [25]. Future research could explore personalized orthopedic interventions such as insoles or corrective footwear designed to mitigate the effects of excessive backpack load. Longitudinal studies are also warranted to determine whether load-induced gait asymmetries contribute to musculoskeletal alterations during growth.

## 5. Conclusions

In summary, external school backpack load significantly impacts most of the dynamic gait parameters analyzed. The findings emphasize the importance of adhering to the recommended schoolbag weight limits and monitoring load exposure in early school-aged children to prevent musculoskeletal disorders. 

This study has several limitations. Firstly, the relatively small sample size may limit the generalizability of the results. Secondly, the analysis was limited to barefoot walking in a controlled laboratory setting, which may not fully reflect real-world school conditions, such as walking on uneven surfaces or wearing different types of footwear. Lastly, although the sample was well-balanced in terms of sex, other variables such as physical activity levels, musculoskeletal maturity, or backpack design were not controlled. The majority of statistically significant differences were observed under 15% and 20% BW loading, suggesting that the school backpack weight for 7-year-old children should not exceed 10% of their body weight. Further research is warranted to assess the mechanisms of postural adaptation to both the placement and magnitude of school backpack load. These efforts should extend beyond research and analysis to include preventive education aimed at children, adolescents, parents, and educators. 

## Figures and Tables

**Figure 2 sensors-25-04219-f002:**
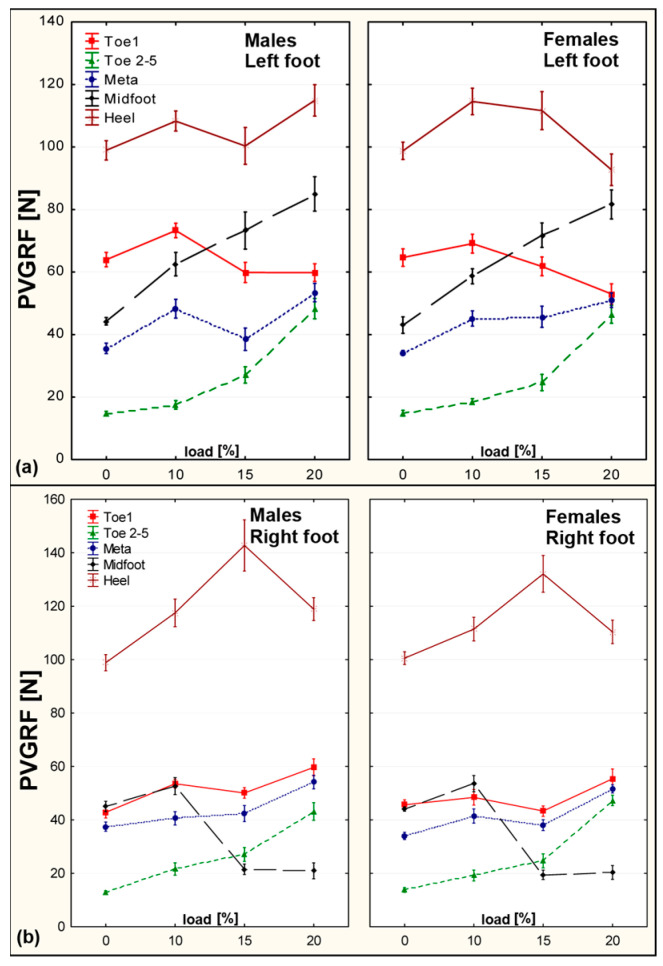
FMAX values across five foot zones for right and left foot—comparison between males and females under different loading conditions. Error bars represent standard error. (**a**) Left foot. (**b**) Right foot.

**Figure 3 sensors-25-04219-f003:**
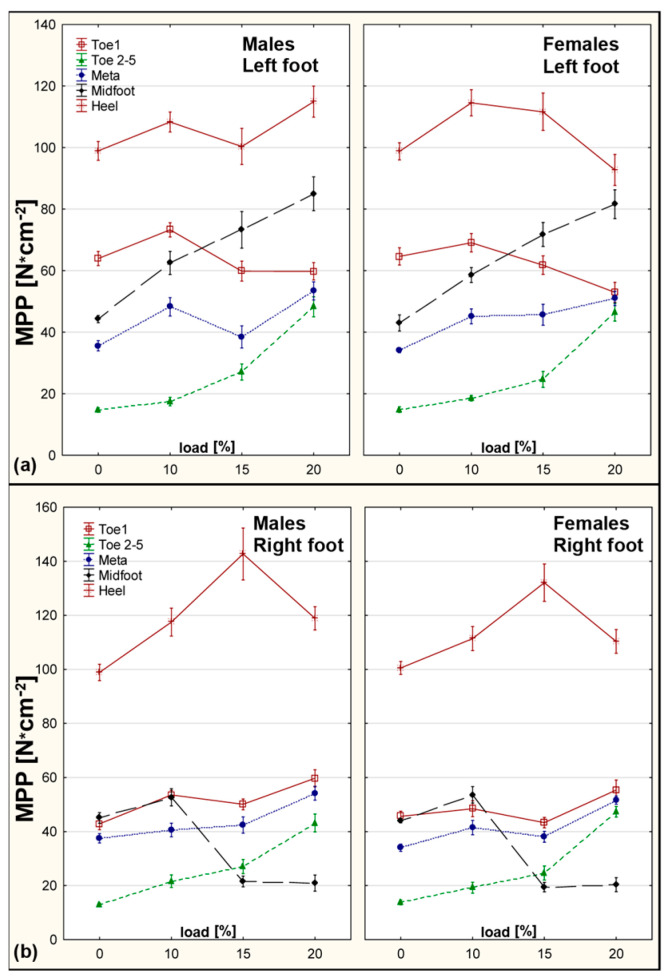
PMAX values for right and left foot—males and females across all loading conditions. (**a**) Left foot. (**b**) Right foot.

**Figure 4 sensors-25-04219-f004:**
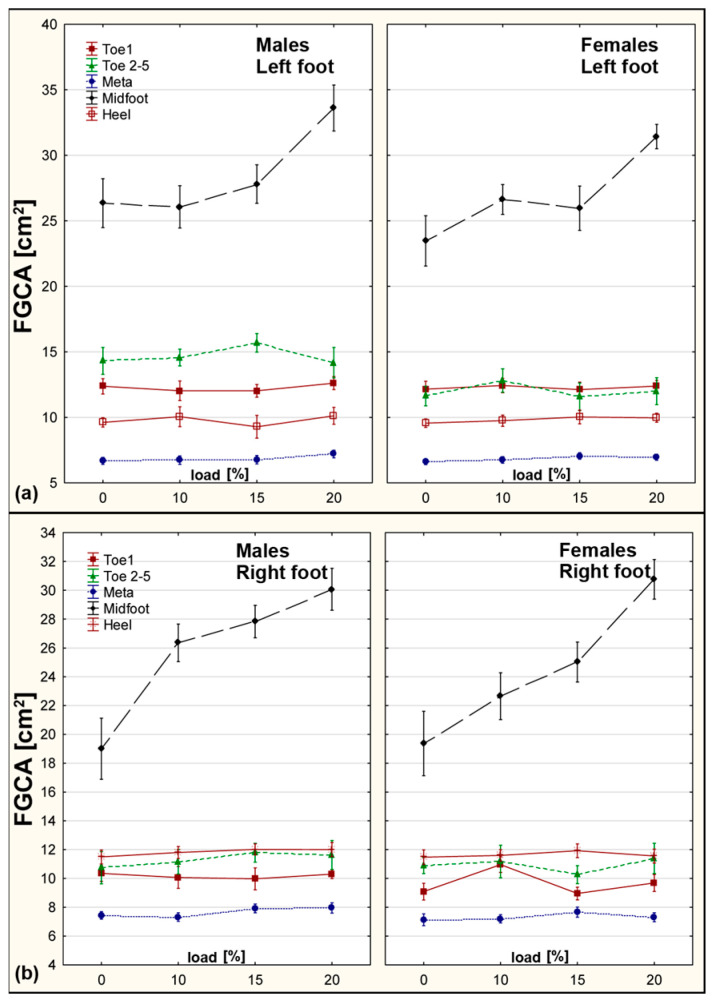
CA values for right and left foot across five zones under increasing load conditions. (**a**) Left foot. (**b**) Right foot.

**Figure 5 sensors-25-04219-f005:**
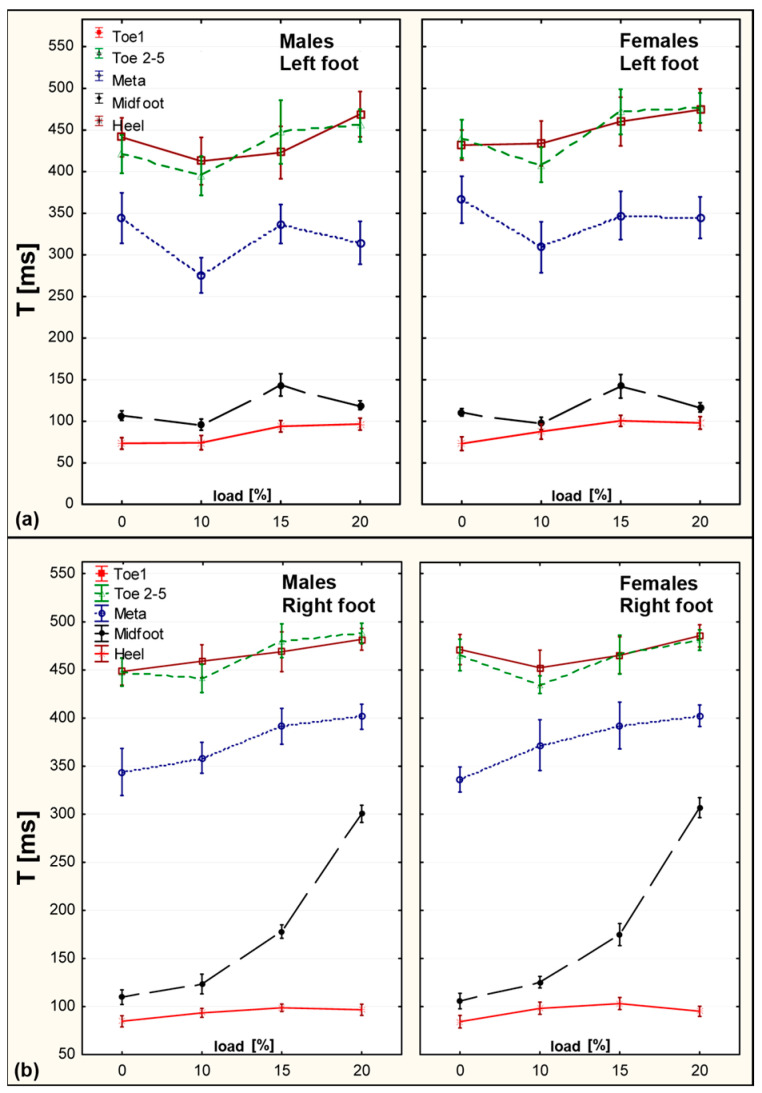
Time to FMAX values for right and left foot—across five zones and four loading conditions. (**a**) Left foot. (**b**) Right foot.

**Table 1 sensors-25-04219-t001:** Basic anthropometric characteristics of the study group (mean ± SD).

Sex	Number of Participants	Body Mass m [kg] AV; SD; CV	Body Height H [cm] AV; SD; CV	BMI [kg/m^2^] AV; SD; CV	Backpack Mass [kg] AV; SD; CV	Backpack Mass – %BW [%] AV; SD; CV
B (Male- Boys)	13	26.49 (±7.81); 0.24	125.54 (±7.3); 0.06	16.55 (±2.89); 0.15	2.08 (±0.32); 0.15	7.09 (±3.7); 0.52
G (Female- Girls)	13	29.19 (±6.89); 0.29	127.46 (±7.1); 0.06	17.77 (±2.68); 0.17	2.41 (±0.64); 0.26	8.42 (±1.72); 0.20

Abbreviations: B—Male (Boys); G—Female (Girls); SD—Standard deviation; AV—arithmetic mean; CV—coefficient of variation, expressed as a ratio of the standard deviation to the mean.

## Data Availability

The data are not publicly available due to data privacy regulations. The data presented in this study are available on request from the corresponding author.

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
