# Peer review of "The Influence of School Backpack Load on Dynamic Gait Parameters in 7-Year-Old Boys and Girls"

_sensors, 2025, doi:10.3390/s25134219_

Round 1
Reviewer 1 Report
Comments and Suggestions for Authors In this paper external school backpack load significantly impacts most of the dynamic gait
parameters analyzed. The authors conclude that the majority of statistically significant differences were observed under 15% and 20% BW loading, suggesting that school backpack weight for 7-year-old children should not exceed 10% of body weight.
My only comment is for the description of the children. It would be interesting to know whether they practice football.

Author Response
Dear Reviewer,
Thank you very much for your thoughtful and constructive feedback on our manuscript.
We would like to address the points raised:
- On originality: While we acknowledge that similar studies have been conducted previously, including our earlier publication in IJERPH (2022), the current manuscript expands the scope significantly by incorporating kinetic parameters, detailed plantar pressure distribution, and symmetry analysis — which were not included in our previous study. Our aim was to provide a more comprehensive understanding of how external load affects pediatric gait mechanics, not limited to spatiotemporal measures.
- On methodology: We appreciate your note that the methodological design is sound. While we did not change the sample, our current analysis involves new data dimensions (including force-based measures and timing of peak values), which justify a separate and focused presentation.
- On physical activity (football practice): This is an excellent suggestion. We confirm that the children included in the study were not professionally involved in sports training or competitive football. A statement has been added in the Methods section for clarification.
We are grateful for your insights and will incorporate the suggested clarification into the revised version.
With kind regards,
Paulina Tomal
on behalf of the authorship team
Reviewer 2 Report
Comments and Suggestions for Authors
Hello, dear colleagues!
The presented work is devoted to a relevant issue — the influence of the weight of school backpacks on gait parameters in young children.
The study is distinguished by a carefully thought-out methodology: a sample of an equal number of boys and girls allows us to compare gender aspects, and the use of the Footscan® system ensures the objectivity and accuracy of measurements. Evaluation of several dynamic parameters of the foot at once (FMAX, PMAX, contact area, time to peak effort) in the sole zone gives a comprehensive picture of the changes taking place. The revealed relationship between increased load and negative changes in gait deserves special attention: increased pressure, increased contact area, asymmetry and slower peak effort time clearly confirm the risk of overloading the musculoskeletal system in young schoolchildren. The authors correctly substantiate the recommendation to adhere to a limit of 10% of body weight — this is an important conclusion for teachers and parents.
A few questions with comments need to be asked about your work
1. How was the sample size calculated?
2. In the tables, "Gender" should be replaced with male and female, since boy and girl are age characteristics of gender.
3. It would be appropriate if you disclose in the introduction statistics on pathology of the musculoskeletal system in children and a possible connection with the load in general.
Within the framework of the topic under discussion, the list of references looks thorough
Author Response
Dear Reviewer,
We would like to express our sincere gratitude for your thoughtful and encouraging feedback on our manuscript. We appreciate your recognition of the methodological rigor and the relevance of our findings, as well as your constructive suggestions for improvement. Please find our responses to your comments below:
- How was the sample size calculated?
Thank you for this important remark. The sample size was determined based on prior studies assessing pediatric gait parameters under load, taking into account an effect size of 0.4, a statistical power of 0.8, and an alpha level of 0.05. This estimation was made using G*Power software, which indicated a minimum of 24 participants. We increased the sample size to 26 (13 girls and 13 boys) to ensure balanced group representation and improved statistical reliability. We have now included this explanation in the Materials and Methods section.
- In the tables, "Gender" should be replaced with male and female, since boy and girl are age characteristics of gender.
Thank you for pointing this out. We agree with your observation and have updated the table headers accordingly by replacing "boys" and "girls" with "male" and "female" to reflect biological sex terminology more accurately.
- It would be appropriate if you disclose in the introduction statistics on pathology of the musculoskeletal system in children and a possible connection with the load in general.
This is a valuable suggestion. We have expanded the Introduction to include current epidemiological data regarding musculoskeletal disorders in school-aged children, highlighting the potential contribution of excessive backpack loads to posture and gait disturbances. This addition strengthens the rationale and significance of our study.
- We fully acknowledge your remarks regarding linguistic clarity. While we have made considerable efforts to ensure the manuscript is written in clear and precise English, we also understand that some editorial improvements may still be needed. We kindly ask for your understanding in this matter. Based on our previous experience with Sensors, we are aware that the handling editor may introduce final language and style corrections during the copyediting stage. We remain committed to making all necessary adjustments to improve the manuscript's quality and readability.
Once again, we thank you for your time, expertise, and constructive guidance. We believe the revisions made in response to your comments have significantly improved the manuscript.
With kind regards,
Paulina Tomal
on behalf of the authorship team
Reviewer 3 Report
Comments and Suggestions for Authors
This paper utilized Footscan system to evaluate the impact of different backpack loads (10%, 15%, and 20% of body weight) on dynamic gait parameters in 7-year-old girls and boys. The key gait parameters were analyzed. The methodology is thorough and the logic flow is clear. The manuscript could be modified before further consideration:
- The introduction part should be emphasized the previous work in this field and the gap between this work and previous work.
- In line 125, the equation should be reformed as the regular form of a equation.
- The limitation of this study is suggested to add in the end of discussion part.
Author Response
Dear Reviewer,
Thank you for your constructive feedback and positive evaluation of our manuscript. We appreciate your valuable suggestions, which have helped us improve the quality of the paper. Please find our responses below:
- Introduction improvement: We have expanded the Introduction section to include a more detailed overview of previous studies in this field and have clarified the specific gap addressed by our work, focusing on the integration of kinetic and plantar pressure parameters in a pediatric population using the Footscan® system.
- Equation formatting (Line 125): The equation has been reformatted to comply with scientific standards. It is now presented in a centered and properly labeled mathematical form.
- Study limitations: A dedicated paragraph outlining the limitations of the study has been added at the end of the Discussion This includes considerations related to sample size, ecological validity, and unmeasured variables.
Thank you once again for your insightful comments.
With kind regards,
Paulina Tomal
on behalf of the authorship team
Reviewer 4 Report
Comments and Suggestions for Authors
The research and results presented in this publication build upon a previous study conducted with the same target group (children aged seven, including 13 boys and 13 girls) regarding the impact of school backpack weight on gait and lower extremity load. The earlier study, published by the same authors three years ago, focused on key temporal and spatial gait parameters, including step length and stride length, heel-to-heel base of support, step time, gait cycle time, single support time, double support time, and walking speed. The current study shifts its emphasis to dynamic gait parameters derived from foot pressure force measured using a "Footscan" sensor platform (specifically, peak vertical ground reaction force, maximum plantar pressure, foot-ground contact area, and time to peak force).
Overall, the results confirm findings from similar research, emphasizing the recommendation that school backpack weight should not exceed 10% of a child's body weight. A more precise analysis concerning the distribution of load in five zones of the foot, the differences between boys and girls, and the asymmetry between the left and right foot may be of scientific interest specifically for this age group; however, it can hardly predict trends regarding subsequent anthropological changes as children grow. A practical application of these findings might involve designing orthopedic insoles or footwear aimed at correcting gait abnormalities caused by loads exceeding 10% of body weight. However, the authors do not discuss these future research directions.
The abstract accurately reflects the article’s content, and reference sources are appropriately cited throughout. Nevertheless, the literature review should be expanded to include relevant studies published within the last 5–10 years.
There are no substantial comments on the content, just a few notes:
- Parameters AV and CV in Table 1 are not discussed.
- The formula used to calculate SI (line 125) is not clear.
- Certain parts of the Discussion, specifically lines 188–196, 234–236, and 281–285, would be more appropriately placed in the Introduction.
Author Response
Dear Reviewer,
Thank you for your thorough and thoughtful review of our manuscript. We greatly appreciate your constructive comments. Below are our responses and corresponding modifications:
- Literature review: We have expanded the Introduction section by adding relevant studies published within the last 5–10 years to better contextualize our research within current scientific discourse.
- Clarification of AV and CV: We added definitions of AV (arithmetic mean) and CV (coefficient of variation) in the table legend to improve clarity.
- Symmetry Index (SI) formula: The formula for calculating the Symmetry Index has been revised and formatted according to standard scientific conventions.
- Discussion restructuring: The indicated lines (188–196, 234–236, and 281–285) have been moved to the Introduction section, where they more appropriately establish the rationale for the study.
- Future directions: We have added a dedicated paragraph at the end of the Discussion section outlining potential future applications of the results, including orthopedic footwear design and longitudinal studies on musculoskeletal development.
Thank you once again for your insightful feedback.
With kind regards,
Paulina Tomal
on behalf of the authorship team